# LUCID-3D: A Lightweight and Compatible Framework for Unified 3D Understanding and Generation

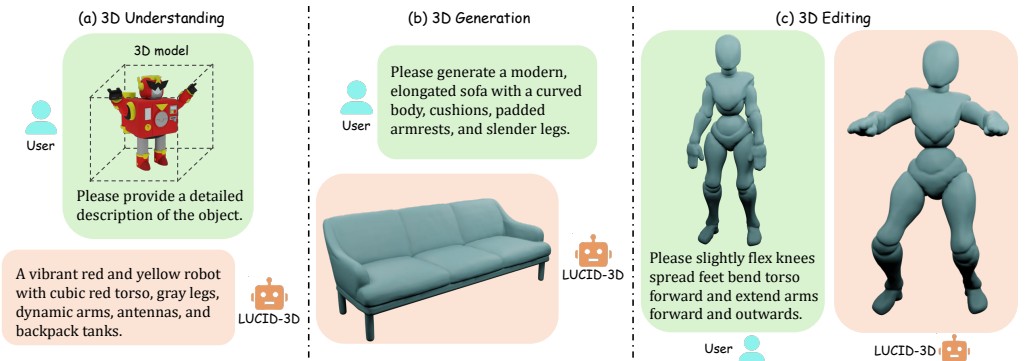

Figure 1: LUCID-3D: the first unified 3D AR+Diffusion framework, enabling 3D understanding, generation, and editing.

## Abstract

The rapid progress of large multimodal models has inspired efforts toward unified frameworks that couple understanding and generation. While such paradigms have shown remarkable success in 2D, extending them to 3D remains largely underexplored. Existing attempts to unify 3D tasks under a single autoregressive (AR) paradigm lead to significant performance degradation due to forced signal quantization and prohibitive training cost. Our key insight is that the essential challenge lies not in enforcing a unified autoregressive paradigm, but in enabling effective information interaction between generation and understanding while minimally compromising their inherent capabilities and leveraging pretrained models to reduce training cost. Guided by this perspective, we present the first unified framework for 3D understanding and generation that combines autoregression with diffusion. Specifically, we adopt an autoregressive next-token prediction paradigm for 3D understanding, and a continuous diffusion paradigm for 3D generation. A lightweight transformer bridges the feature space of large language models and the conditional space of 3D diffusion models, enabling effective cross-modal information exchange while preserving the priors learned by standalone models. Extensive experiments demonstrate that our framework achieves state-of-the-art performance across diverse 3D understanding and generation benchmarks, while also excelling in 3D editing tasks. These results highlight the potential of unified AR+diffusion models as a promising direction for building more general-purpose 3D intelligence. Our code and models will be released.

## 1 Introduction

Recently, with the advent of GPT-4o (Hurst et al., 2024) and its remarkable performance in 2D understanding and generation, the research community has made substantial progress in exploring unified 2D models (Chen et al., 2025a; Deng et al., 2025; Pan et al., 2025; Ma et al., 2025), particularly in terms of data representation, model architectures, and training strategies. Such unification

promises synergistic benefits: a single model can share representations between perception and synthesis, improve data efficiency, and enable new capabilities like multimodal reasoning. In parallel, both 3D understanding (Xu et al., 2024) and 3D generation (Zhang et al., 2023; Xiang et al., 2025; Zhang et al., 2024; Lei et al., 2025) have respectively reached new heights. Nevertheless, unified models that jointly address 3D understanding and generation remain largely underexplored.

Inspired by recent successes of unified understanding–generation models in 2D, we aim to develop a unified framework for 3D understanding and generation, taking a pivotal step toward more capable and versatile 3D models. The central challenge is to bridge the asymmetry between large-scale language priors for understanding and the comparatively small 3D datasets for generation, enabling information exchange while preserving the strengths of both. ShapeLLM-Omni (Ye et al., 2025) recently pioneered a purely autoregressive (AR) framework to unify 3D understanding and generation. While this approach aligns training objectives under a single paradigm, it requires discretizing continuous 3D signals into tokens, which inevitably degrades performance and prevents effective use of pretrained LLM priors. Consequently, it suffers from information loss due to quantization, high training costs, and limited compatibility with existing models and pipelines.

In this work, we present, to our knowledge, the first unified 3D AR+diffusion framework that couples autoregression for understanding with diffusion for generation. The framework is lightweight, broadly compatible, and better preserves priors learned by specialized understanding and generation models. This design is particularly suitable for 3D, where data is far scarcer than in 2D images-text corpora. Our insight is that the key to this task lies not in enforcing a unified autoregressive paradigm but in *enabling efficient interaction between 3D understanding and generation while preserving their respective strengths and leveraging pretrained models*.

Our framework employs an autoregressive next-token prediction paradigm for the understanding, while the generation follows the continuous diffusion paradigm. Concretely, we append a fixed-length set of learnable query tokens to the user input, which elicit relevant knowledge and produce informative representations via the language model (Bai et al., 2025). A lightweight transformer then projects these representations into the original conditional embedding space of a 3D diffusion model, serving as an adaptation module between the LLM feature space and the diffusion generator. This preserves the native AR and diffusion paradigms while enabling effective information exchange without degrading 3D signal fidelity. The same mechanism naturally extends to 3D editing: providing 3D inputs to the understanding module yields guidance signals that pilots generation according to user instructions. The framework is computationally efficient and integrating seamlessly with diverse pretrained 3D understanding and generation models.

Extensive quantitative and qualitative experiments, along with ablations, demonstrate strong performance on both 3D understanding and generation tasks and validate the effectiveness of the lightweight design. We also showcase promising results on 3D editing.

## 2 RELATED WORK

**3D Generative Models.** Previous methods for 3D generation can be categorized into two families. The first approach leverages 2D diffusion priors Song et al. (2021); Ho et al. (2020); Esser et al. (2021) to generate 3D objects via an optimization-based pipeline (Poole et al., 2023; Wang et al., 2023; Chen et al., 2023). However, these methods are computationally intensive and prone to oversaturation artifacts. In contrary, native 3D generative models (Zhang et al., 2023; Vahdat et al., 2022; Lan et al., 2024; Li et al., 2025b; Lan et al., 2025; Xiang et al., 2025) have recently emerged for high-quality, efficient, and scalable 3D generation. Most follow a two-stage pipeline: (i) learning a VAE (Kingma & Welling, 2013; Tian et al., 2024) that encodes 3D objects into a latent space (Zhang et al., 2023; Cho et al., 2025; Chen et al., 2025b;c), and (ii) training a latent generative model on these latents (Peebles & Xie, 2023; Ma et al., 2024; Tian et al., 2024). Owing to their scalability, such native 3D generative models are already being explored for commercial applications (Hunyuan3D et al., 2025; Zhang et al., 2024; Wu et al., 2024b). However, despite this progress, their editing capabilities—especially under a unified understanding–and–generation paradigm (Hurst et al., 2024)—remain underexplored. What is more, our method demonstrates remarkable 3D generation capability, as shown in Figure 2.

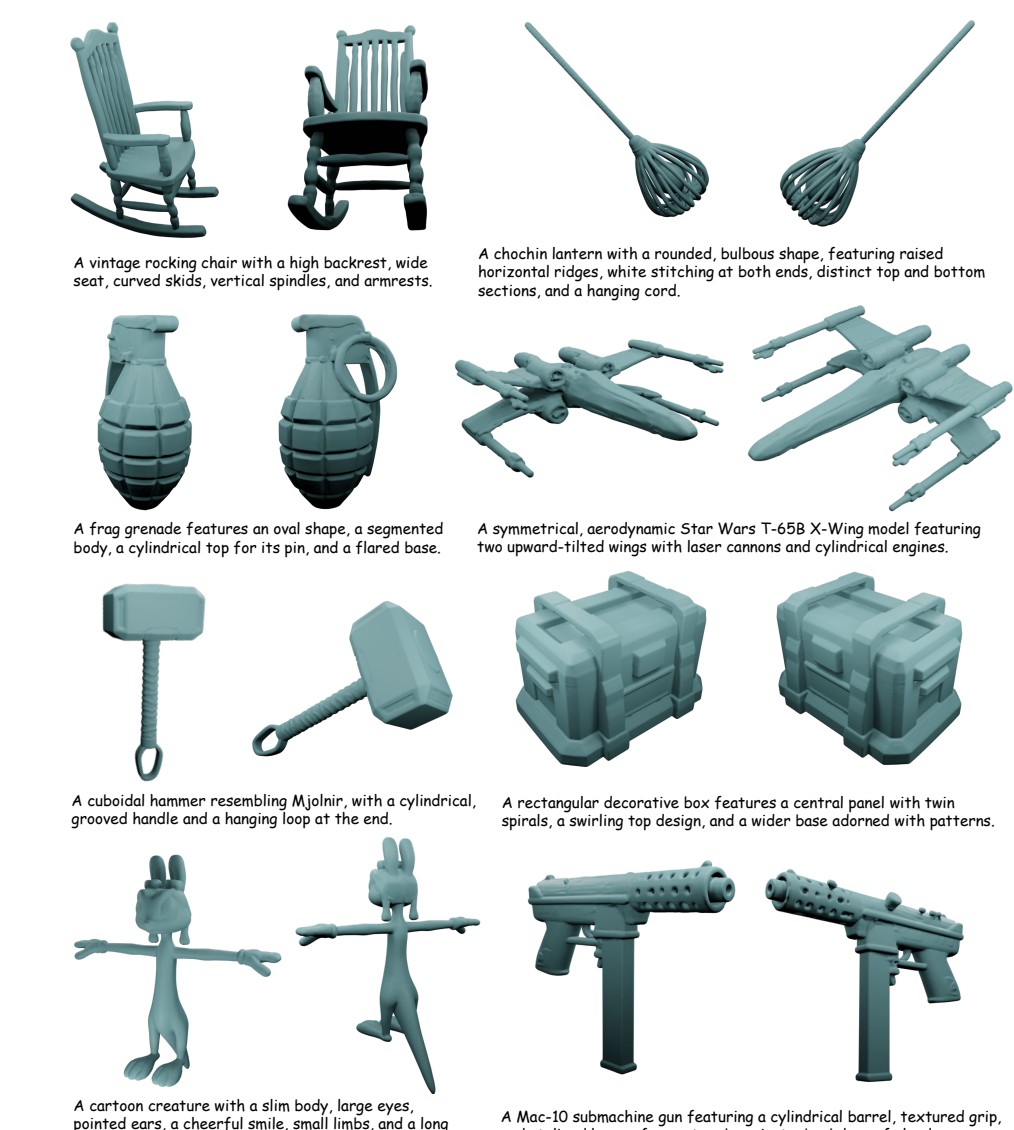

Figure 2: Qualitative text-to-3D generation results. LUCID-3D is capable of generating diverse and high-quality 3D models.

**3D Understanding Models.** The rapid progress of large language models (LLMs) (Achiam et al., 2023; Touvron et al., 2023a;b; Yang et al., 2025) has catalyzed multimodal research that grounds language in 3D signals. Recent 3D understanding models integrate point clouds, depths, and multi-view images to support tasks such as 3D captioning, question answering, grounding, and reasoning about geometry and semantics (Luo et al., 2023; 2024; Zhou et al., 2024b; Hong et al., 2023a;b; Qi et al., 2024; Xu et al., 2024; Zhou et al., 2025). Despite steady gains, these systems are designed for understanding and therefore do not natively support 3D generation or editing.

**Unified Understanding and Generation Models.** Pioneered in 2D by GPT-4o (Hurst et al., 2024), unified models have driven advances in data curation (Chen et al., 2025a; Deng et al., 2025; Liu et al., 2024), representation design (Zhao et al., 2025; Lin et al., 2025; Song et al., 2025), architectures (Xie et al., 2024; Wu et al., 2025; Chen et al., 2025a), and training strategies (Deng et al., 2025; Zhuang et al., 2025b; Li et al., 2025a). Representative paradigms include purely autoregressive (AR) (Wu et al., 2024c;a; Wang et al., 2024), AR+diffusion (Zhou et al., 2024a; Chen et al., 2025a), and AR+visual-autoregression (VAR) (Zhuang et al., 2025a). On the 3D side, recently ShapeLLM-Omni (Ye et al., 2025) proposed a purely AR pipeline; while objective-unified, it suffers from quantization-induced information loss, high training cost, and limited compatibility with ex-

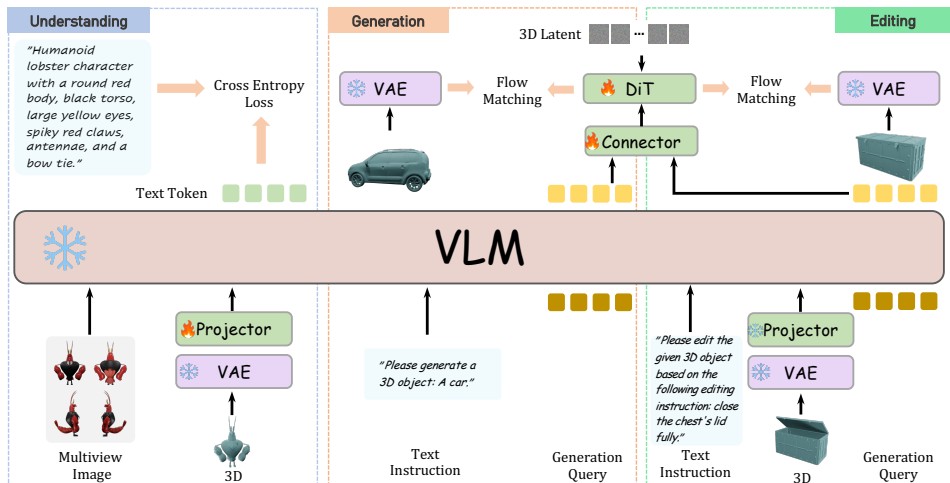

Figure 3: Framework Overview. LUCID-3D uses an autoregressive next-token paradigm for understanding and a continuous diffusion paradigm for generation. The two components interact via a trainable cross-modal connector.

isting 3D generative backbones. In this work, we introduce (to our knowledge) the first unified 3D AR+Diffusion framework: *autoregression for understanding, diffusion for generation*. The design is lightweight, highly compatible, and preserves strong priors from standalone understanding and generation models—properties especially valuable in the data-scarce 3D regime.

## 3 PROPOSED METHOD

In this section, we present LUCID-3D, a lightweight framework for unified 3D understanding and generation that integrates the pretrained priors from an autoregressive vision-language model (VLM) and a 3D diffusion model. More formally, we aim to approximate the joint distribution $p_\theta(x, t)$ over 3D shapes $x$ and text $t$, enabling inference over its conditionals for understanding $p_\theta(t \mid x)$, generation $p_\theta(x \mid t)$, and editing $p_\theta(x_{\text{edit}} \mid x_{\text{input}}, t)$. Practically, the system accepts text prompts and/or 3D shapes and produces semantic outputs or generated/edited 3D content.

Figure 3 overviews the pipeline. Section 3.1 describes how we extend a 2D VLM to operate 3D latents via a lightweight projector while keeping the VLM frozen. Section. 3.2 details the 3D generation module, where learnable queries extract knowledge from the VLM and a transformer connector aligns these representations to the conditional space of a pretrained 3D diffusion model. Section 3.3 shows how the same mechanism supports instruction-driven 3D editing conditioned on an input shape. Finally, Section 3.4 outlines data curation.

### 3.1 3D UNDERSTANDING

We extend a 2D VLM to the 3D domain to model $p_\theta(t \mid x)$, where $x$ is a 3D shape and $t$ is text. We use QwenVL-2.5 (Bai et al., 2025) as the base VLM $\Phi$ and encode $x$ with the Hunyuan3D 2.1 VAE (Hunyuan3D et al., 2025) denoted $\Omega$, yielding a sequence of continuous 3D latents $u = \Omega(x)$. To interface with $\Phi$, we introduce a lightweight MLP projector $\varphi$ (four fully connected layers) that maps $u$ to the VLM's token embedding space. Training follows standard autoregressive next-token prediction with cross-entropy:

$$\hat{t} = \Phi\big(\varphi(\Omega(x))\big), \quad \mathcal{L}_{\text{und}} = \text{CE}\big(\hat{t}, t_{\text{gt}}\big), \tag{1}$$

where $\hat{t}$ denotes the predicted token distribution and $t_{\text{gt}}$ the ground-truth text. We freeze $\Phi$ (and $\Omega$) and train only $\varphi$. This preserves the VLM's general multimodal priors while enabling 3D understanding.

At inference, users may optionally provide multi-view images to supply texture cues missing from the 3D latents. These images are processed by the VLM's vision tower. We use the following prompt template: *"Given <ThreeD> the 3D representation and <image> multi-view images of the object (front, back, left, right), provide a detailed description of the object."*

## 3.2 3D Generation

We next describe the text-to-3D generation pipeline, which approximates the conditional distribution $p_\theta(x \mid t)$, where $t$ is a user-provided textual instruction and $x$ is the generated 3D shape. Inspired by (Chen et al., 2025a; Pan et al., 2025), given input text $t$, we concatenate a fixed-length set of learnable queries $Q$ after $t$ to form the sequence $[t; Q]$. Passing it through the VLM $\Phi$ yields enriched high-level features in which the queries capture domain knowledge and context from the pretrained model. These query features serve as a cross-modal bridge to the 3D generation module.

To align the query features with the conditional input space of the pretrained 3D diffusion model, we employ a transformer connector $\omega$. This connector maps the sequence $\Phi([t; Q])$ into the conditioning space expected by the diffusion transformer DiT, which is pretrained for image-conditioned 3D generation. The forward process is:

$$z = \text{DiT}\big(\omega(\Phi([t; Q])),\, \epsilon\big), \quad \mathcal{L}_{\text{gen}} = \text{FM}\big(\Omega(x_{\text{gt}}),\, z\big), \tag{2}$$

where $z$ is the generated 3D latent, $\epsilon$ is Gaussian noise, $\Omega(x_{\text{gt}})$ is the ground-truth latent encoding of the target 3D shape, and FM denotes the flow-matching loss (Ma et al., 2024; Lipman et al., 2023; Liu et al., 2023). By supervising with $\mathcal{L}_{\text{gen}}$, the model learns to align the diffusion outputs with the ground-truth latent codes. During training, we freeze the VLM $\Phi$ and optimize the queries $Q$, connector $\omega$, and diffusion backbone DiT. This ensures efficient training while leveraging powerful pretrained priors for both language and 3D generation.

## 3.3 3D Editing

Finally, we extend the framework to instruction-guided 3D editing. The objective is to approximate $p_\theta(x_{\text{edit}} \mid x_{\text{input}}, t)$, where $x_{\text{input}}$ is an input 3D shape, $t$ is a textual instruction, and $x_{\text{edit}}$ is the resulting edited shape.

We begin by encoding $x_{\text{input}}$ into a latent $\Omega(x_{\text{input}})$. This latent, together with the text $t$ and learnable queries $Q$, forms the combined sequence $[t; \Omega(x_{\text{input}}); Q]$. This sequence is processed by the VLM $\Phi$ to extract features that jointly encode both the 3D structure of the input and the textual guidance. The connector $\omega$ maps these features to the conditional space of the pretrained 3D diffusion model DiT, which then produces an edited latent $z_{\text{edit}}$:

$$z_{\text{edit}} = \text{DiT}\big(\omega\big(\Phi([t; \Omega(x_{\text{input}}); Q])\big),\, \epsilon\big), \qquad \mathcal{L}_{\text{edit}} = \text{FM}\big(\Omega(x_{\text{edit}}),\, z_{\text{edit}}\big). \tag{3}$$

Here $\mathcal{L}_{\text{edit}}$ supervises the alignment of the predicted edited latent with the ground-truth edited latent $\Omega(x_{\text{edit}})$. This setup leverages the VLM's understanding capability to condition generation on both shape and text, producing edits that respect the geometry of the original while reflecting the instruction.

## 3.4 Dataset

We curate two datasets: a *Text–3D* dataset pairing shapes with multi-granularity textual descriptions for understanding and generation, and a *3D Editing* dataset pairing edited shapes with prompts. Illustrations are provided in Appendix A.3.1.

**Text–3D Dataset.** For each 3D mesh, we render four canonical views (front, back, left, right) under fixed camera intrinsics and arrange them into a $2 \times 2$ multi-view grid. This grid is provided to GPT-4o to produce multi-granularity 3D captions. Each caption set covers: (i) semantic category, (ii) geometric structure (global shape and part-level attributes), (iii) texture/material appearance, and (iv) function and style descriptors. We perform quality control via self-consistency checks across granularities and remove near-duplicate or low-content captions. To expose the model to varying levels of user specificity, we provide 10 annotation levels ranging from concise to detailed, with examples in Fig. 10.

**3D Editing Dataset.** To preserve 3D consistency in edit pairs, we leverage 4D sequences (animated objects) and treat different frames as pre-/post-edit states. Starting from a filtered list of 4D objects (Liang et al., 2024), we render four canonical views (front, back, left, right) of both the first and the $24^{\text{th}}$ frames. The four views of each frame are concatenated into a $2 \times 2$ grid, yielding an image pair per object. We prompt a vision–language model (Gemini (Team, 2025)) to (a) retain only pairs that exhibit a salient object-centric change (excluding camera/lighting changes), and (b)

Table 1: 3D object captioning results on Objaverse (Deitke et al., 2023b;a). Following PointLLM (Xu et al., 2024), we note that BLEU-1, ROUGE-L, and METEOR often favor short captions and insufficiently reflect semantic accuracy or diversity. Accordingly, we highlight Sentence-BERT and SimCSE scores as more reliable metrics.

| Method | BLEU-1 | ROUGE-L | METEOR | Sentence-BERT | SimCSE |
|---|---|---|---|---|---|
| PointLLM-7B (Xu et al., 2024) | 8.70 | 11.07 | 13.25 | 55.76 | 58.27 |
| PointLLM-13B (Xu et al., 2024) | 8.53 | 11.00 | 13.08 | 55.50 | 57.96 |
| ShapeLLM-Omni (Ye et al., 2025) | 13.55 | 16.44 | 10.50 | 35.71 | 38.66 |
| QwenVL-2.5 (2D) | 8.04 | 8.33 | 6.27 | 42.75 | 47.93 |
| **Ours (3D)** | 12.86 | 15.38 | 10.26 | 45.80 | 47.42 |
| **Ours (2D+3D)** | **19.75** | **19.97** | **16.41** | **61.03** | **64.80** |

generate a concise editing instruction describing the transformation. The detailed instruction template used in the API call is provided in the Appendix A.3.2. For data augmentation, we request two instructions per retained pair: one for the forward change (frame 1 $\rightarrow$ frame 24) and one for the reverse (frame 24 $\rightarrow$ frame 1). This procedure yields over 14K 3D shape–editing-prompt pairs.

# 4 EXPERIMENTS

## 4.1 IMPLEMENTATION DETAILS

For the Text-3D dataset, we curate a high-quality corpus from G-Objaverse (Qiu et al., 2024), where each object is annotated with 10 levels of captions ranging from concise to detailed, resulting in over 320K objects with 10 captions each. For the 3D editing dataset, we filter Diffusion4D (Liang et al., 2024) and obtain more than 14K shape–prompt pairs. For 3D understanding, we adopt a high-quality subset of 40K shapes, training only the MLP projector while keeping Qwen2.5 (Bai et al., 2025) as the base model. For the generation task, we initialize the DiT weights from the pretrained image-conditioned Hunyuan 2.1 (Hunyuan3D et al., 2025), and jointly train the DiT and connector with 16 layers and 8 heads. Training is conducted for 280K steps on 24 GPUs with a batch size of 672. For the 3D editing task, we train on 8 GPUs for 80K steps with a batch size of 112. Across both generation and editing, we finetune all parameters of the connector and DiT using the AdamW optimizer with a learning rate of $1 \times 10^{-4}$. We set the 3D latent size to 1024 during training, with a query length of 64 for generation and 1024 for editing. During inference, the 3D latent size is increased to 4096.

## 4.2 QUANTITATIVE AND QUALITATIVE COMPARISONS

**3D Understanding.** We annotated a high-quality set of 917 detailed object captions on the Objaverse dataset (Deitke et al., 2023b) using GPT-4o to serve as ground truth for evaluating different methods. Table 1 and Fig. 4 present both quantitative and qualitative comparisons between our approach and other state-of-the-art methods on the 3D object captioning task. As noted in PointLLM (Xu et al., 2024), BLEU-1, ROUGE-L, and METEOR tend to favor short captions and are limited in capturing semantic accuracy and diversity. We therefore place greater emphasis on Sentence-BERT and SimCSE scores. It is evident that when relying solely on 3D latents, our method may suffer from hallucinations regarding color and other texture-related attributes due to the lack of explicit texture information. Nevertheless, our approach still outperforms ShapeLLM-Omni (Ye et al., 2025). Fig. 4 shows that PointLLM (Xu et al., 2024) tends to produce verbose descriptions with invalid information and hallucinations. Furthermore, by incorporating multi-view 2D images to provide complementary texture cues, our method achieves the best overall performance. This also highlights the strong scalability and flexibility of our framework. We additionally report the performance of our base model QwenVL-2.5 on multi-view images in Table 1 to further demonstrate the superiority of our approach.

**3D Generation.** We evaluate our method on the text-to-3D generation task. Qualitative results are presented in Fig. 2, showing that by incorporating a Transformer-based connector, our approach effectively maps the conditional space from images to high-level text, enabling the generation of detailed 3D shapes that align well with user prompts. This demonstrates the effectiveness of our

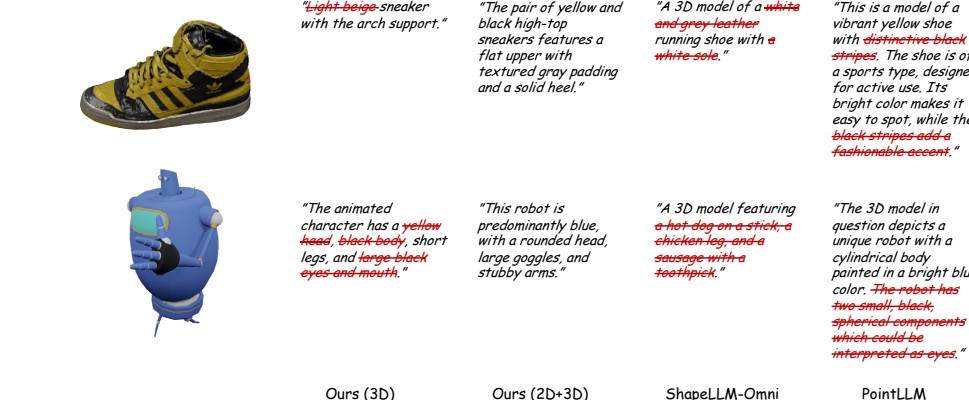

*"~~Light beige~~ sneaker with the arch support."*   *"The pair of yellow and black high-top sneakers features a flat upper with textured gray padding and a solid heel."*   *"A 3D model of a ~~white and grey leather~~ running shoe with ~~a white sole~~."*   *"This is a model of a vibrant yellow shoe with ~~distinctive black stripes~~. The shoe is of a sports type, designed for active use. Its bright color makes it easy to spot, while the ~~black stripes add a fashionable accent~~."*

*"The animated character has a ~~yellow head~~, ~~black body~~, short legs, and ~~large black eyes and mouth~~."*   *"This robot is predominantly blue, with a rounded head, large goggles, and stubby arms."*   *"A 3D model featuring ~~a hot dog on a stick, a chicken leg, and a sausage with a toothpick~~."*   *"The 3D model in question depicts a unique robot with a cylindrical body painted in a bright blue color. ~~The robot has two small, black, spherical components which could be interpreted as eyes~~."*

Ours (3D)          Ours (2D+3D)          ShapeLLM-Omni          PointLLM

Figure 4: Qualitative comparison on 3D object captioning. Benefiting from the flexibility of our framework, combining additional 2D information with the 3D latent yields more accurate and detailed captions. In contrast, PointLLM (Xu et al., 2024) often generates overly long outputs with meaningless redundancy and still suffers from hallucination issues. Note that our method may produce hallucinations due to the lack of texture information in the 3D latent.

Table 2: Quantitative comparison on text-to-3D generation. We report CLIP-score (Radford et al., 2021) (ViT-L/14 and RN50×4), Q-Align (Wu et al., 2023), and MUSIQ-AVA (Ke et al., 2021) across four representative baselines and our method. Our approach achieves the best Q-Align score, indicating superior text–shape alignment, while maintaining competitive performance on CLIP and MUSIQ-AVA.

| Method | ViT-L/14↑ | RN50×4↑ | MUSIQ-AVA↑ | Q-Align↑ |
|---|---|---|---|---|
| Trellis (Xiang et al., 2025) | **27.91** | **42.75** | **5.20** | 2.11 |
| ShapeLLM-Omni (Ye et al., 2025) | 25.63 | 40.74 | 5.05 | 1.81 |
| SAR3D (Chen et al., 2025c) | 25.88 | 40.80 | 4.90 | 1.79 |
| GaussianAnything (Lan et al., 2025) | 24.91 | 38.71 | 4.62 | 1.56 |
| **Ours** | 27.54 | 42.19 | 5.07 | **2.12** |

design. For quantitative evaluation, we use Gemini (Team, 2025) to generate 643 text prompts and compare against four representative text-to-3D baselines: Trellis (Xiang et al., 2025), ShapeLLM-Omni (Ye et al., 2025), SAR3D (Chen et al., 2025c), and GaussianAnything (Lan et al., 2025). Each generated object is rendered into 24 views for evaluation. We report CLIP-score (Radford et al., 2021) with ViT-L/14 and RN50×4 backbones, Q-Align (Wu et al., 2023) to assess alignment between generated assets and text prompts, and MUSIQ-AVA (Ke et al., 2021) to measure aesthetic quality. As shown in Table 2, our method achieves the best Q-Align score, indicating superior text–shape alignment. While Trellis slightly outperforms us on CLIP and MUSIQ-AVA scores, our method still surpasses ShapeLLM-Omni, SAR3D, and GaussianAnything by a clear margin. More importantly, unlike these baselines which are either trained from scratch or finetuned from text-to-3D models, our pipeline is initialized from an image-conditioned model and is further extensible to 3D understanding and editing tasks beyond pure text-to-3D generation. Beyond quantitative results, Fig. 5 highlights that our model can capture fine-grained details explicitly described in the prompt but often ignored by other methods. For example, given the instruction "circular cut-outs", our model faithfully generates the circular cut-out, whereas competing approaches miss this detail. This suggests that the query extracted from the VLM prior enables our model to attend more closely to subtle textual cues, even if such improvements are not fully reflected in standard metrics. In addition, we compare with the base image-conditioned model Hunyuan 2.1 (Hunyuan3D et al., 2025). Specifically, we first generate an image using the input prompt with Flux (Labs, 2024), and then feed the image into Hunyuan 2.1. As shown in Fig. 5, our lightweight design is able to preserve the strong generative ability of the base model while incorporating strong text alignment, thus achieving both high-quality geometry and faithful adherence to user prompts.

**3D Editing.** We further demonstrate the capability of our pipeline on 3D editing. As shown in Fig. 6, the user can provide a 3D shape together with a natural language editing prompt, and our model modifies the overall geometry accordingly. Unlike traditional approaches that require users

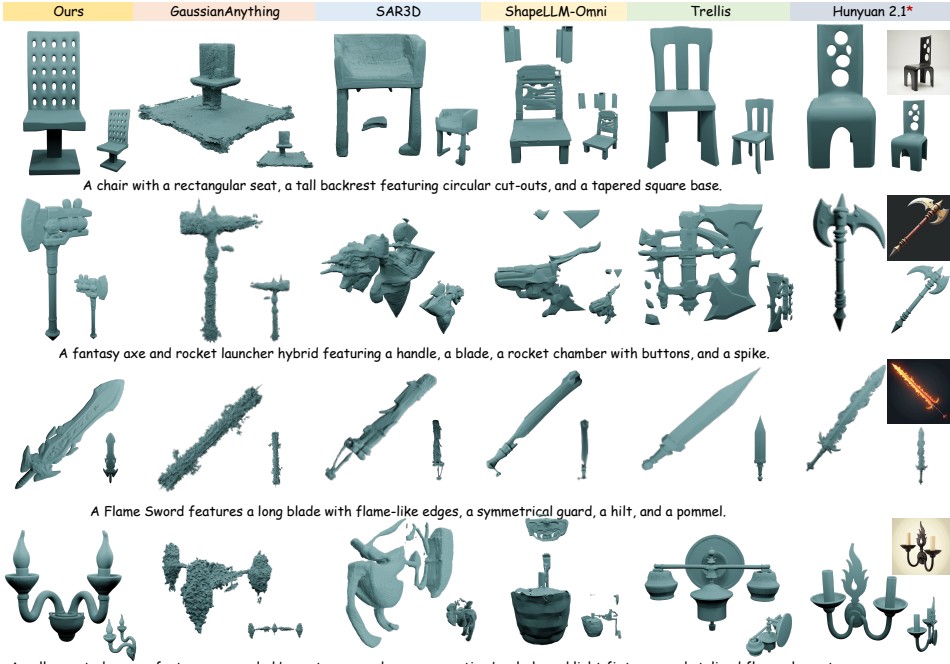

Figure 5: Qualitative comparison on 3D generation. Compared with competing methods, our approach generates high-quality 3D shapes, capturing fine-grained details while aligning well with user input, and meanwhile preserves the generative ability of the base image-conditioned model. * indicates the image-conditioned base model used.

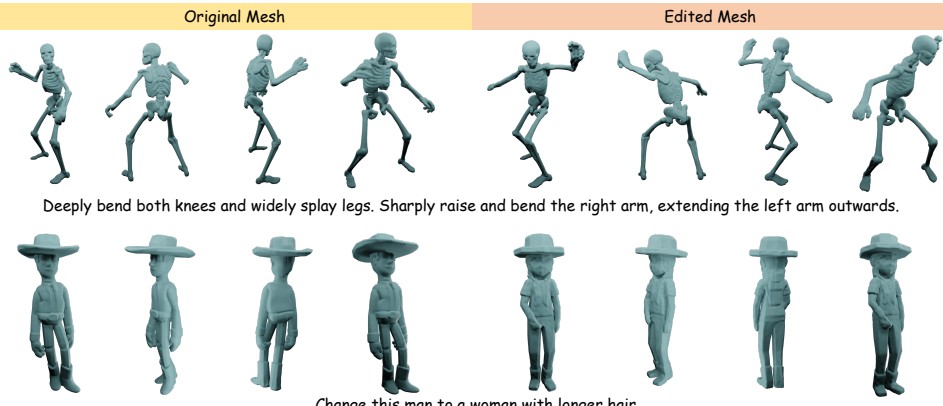

Figure 6: Qualitative results on 3D editing. Our method edits input meshes according to natural language prompts, achieving diverse pose and appearance changes while preserving identity-related attributes, and it also works for out-of-distribution edits.

to manually specify local regions to be edited, our design directly interprets the prompt and 3D shape latent at the semantic level. For example, when asked to "extending the left arm outwards," our model automatically identifies the relevant limbs and performs consistent articulations, while preserving the identity and style of the original shape. To achieve this, we set a longer query length (1024) to capture more fine-grained part-level semantics from the input mesh, which enables the model to better maintain shape identity during editing. Users only need to describe the desired modification in natural language, while the model automatically infers the spatial regions to be edited. Moreover, our method demonstrates the ability to handle prompts that are out of distribution. For instance, as shown in the second example, our model successfully transforms the male figure into a female figure with longer hair, while preserving the original hat, boots, and standing pose. This type of appearance modification lies outside the distribution of our dataset, which predominantly focuses on pose variations and shape deformations.

## 4.3 ABLATION STUDY

**Different Query Length for 3D Editing.** In our editing experiments, we observe that the number of query tokens required by the generation module plays a crucial role in preserving the fidelity of the input 3D object. Fig. 7 illustrates the editing results with token lengths ranging from 128 to 1024. It is evident that longer query tokens effectively alleviate the information bottleneck, thereby better retaining the original attributes of the input 3D object. Moreover, as the token length increases, the edited object's pose aligns more closely with the textual instruction, while the overall shape identity of the input mesh is also better preserved. These results highlight that increasing the token length consistently improves both semantic alignment and geometric consistency in 3D editing.

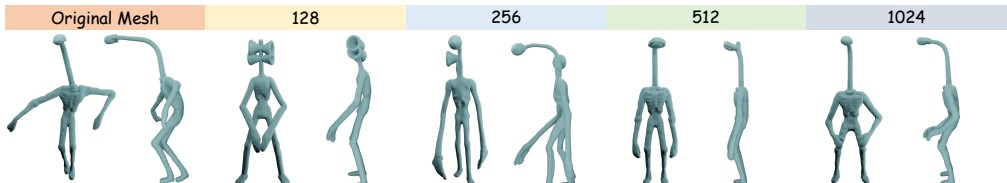

Upright the torso gently straighten the neck fold arms inwards and deeply bend the knees to lower the figure.

Figure 7: Ablation on query length for 3D editing. Editing results with different query token lengths (128–1024). Longer queries alleviate the information bottleneck and better preserve the original attributes of the input mesh while performing the requested edits.

**Robust and Lightweight Training Strategy.** Although our 3D generative model is trained only on 3D latents of length 1024, it generalizes effectively to longer token lengths (2048 and 4096) at inference, producing more fine-grained and detailed 3D shapes (Fig. 8). This highlights not only the strong prior knowledge preserved from the base model (Hunyuan3D et al., 2025) and the compatibility of our framework, but also the efficiency of our training strategy—requiring low training cost while supporting high-quality generation with extended latents.

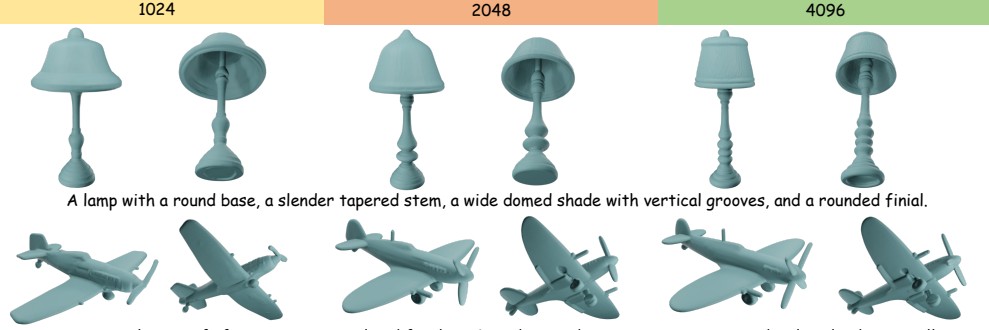

A lamp with a round base, a slender tapered stem, a wide domed shade with vertical grooves, and a rounded finial.

A vintage-style aircraft featuring a streamlined fuselage, broad upward-curving wings, a tapered tail, a slender propeller, minimalist landing gear, and distinct accents.

Figure 8: Generalization to longer latents. Our model is trained with 1024 tokens but supports longer latents at inference, yielding finer 3D details with low training cost.

## 5 CONCLUSIONS

In this work, we presented the first unified framework that integrates autoregression for 3D understanding with diffusion for 3D generation. By disentangling the paradigms of understanding and generation while enabling effective information exchange through a lightweight transformer, our approach preserves the strengths of standalone models and achieves state-of-the-art results across diverse 3D tasks. Moreover, we demonstrated the strong potential of our framework in 3D editing, highlighting its flexibility and extensibility. We believe that our work sheds new light on the development of future unified 3D understanding–generation models and intelligent 3D editing models. We hope it will further inspire the community to explore better architectures, training paradigms, and frameworks for advancing unified 3D intelligence.

# 6 REPRODUCIBILITY STATEMENT

All code and model checkpoints will be publicly released to ensure reproducibility.

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

# A APPENDIX

## A.1 USE OF LARGE LANGUAGE MODELS

Large Language Models (LLMs) were used solely for minor grammar correction and stylistic polishing of the manuscript. They were not involved in the design of the methodology, execution of experiments, analysis of results, or any other aspect of the scientific contribution.

## A.2 ABLATION STUDY

### A.2.1 ADAPTATION TO DIFFERENT 3D GENERATIVE MODEL.

To verify the compatibility of our framework with different 3D generation models, we replace the generation base model with CraftsMan3D (Li et al., 2024) while keeping all other settings unchanged. We train both models on 10K objects for 40K steps with a batch size of 56. As shown in Figure 9, our framework achieves text-to-3D synthesis with low training cost, demonstrating its lightweight and highly compatible nature.

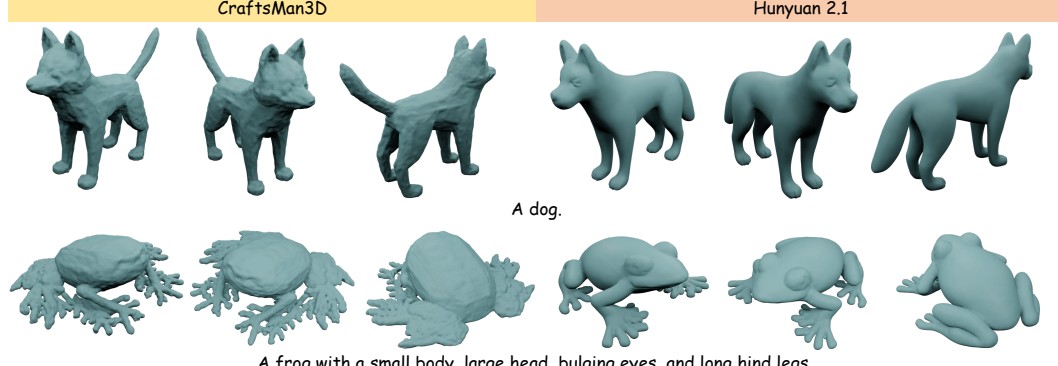

A dog.

A frog with a small body, large head, bulging eyes, and long hind legs.

Figure 9: Adaptation to different 3D generative models. Our framework works with different base models, demonstrating strong compatibility and producing 3D shapes faithful to text prompts.

## A.3 DATASET DETAILS

### A.3.1 ILLUSTRATION OF TEXT-3D AND 3D EDITING DATASETS

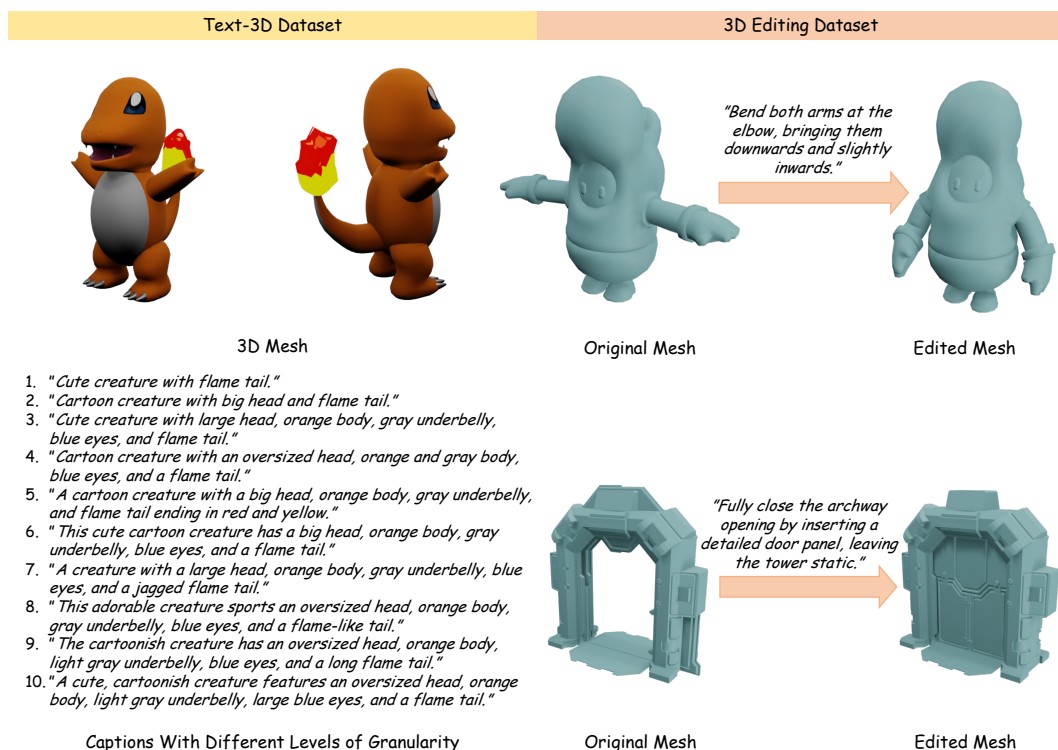

Figure 10: Illustration of our *Text-3D* and *3D Editing Datasets*. The *Text–3D Dataset* pairing 3D shapes with multi-granular textual descriptions for 3D understanding and generation, and the *3D Editing Dataset* pairing edited 3D shapes with editing prompts for instruction-driven editing.

### A.3.2 INSTRUCTION TEMPLATE FOR CONSTRUCTING THE 3D EDITING DATASET

As described in the main text, our 3D editing dataset is constructed by treating different frames of 4D sequences as the pre- and post-editing states. Each frame is rendered from the four canonical views (front, back, left, and right) and stitched into a $2 \times 2$ grid multi-view image. We then query the Gemini API (Team, 2025) to generate the corresponding editing instructions. Specifically, the template used to query Gemini are as follows:

```
Act as a meticulous 3D technical artist. Your task is to analyze two
multi-view composite images ('before' and 'after'). Each image is
stitched together from four perspectives (front, back, left, and right)
to show the entire 3D object.

Based on your analysis, generate a single, descriptive instruction that
accurately explains the transformation.

Your instruction must follow two strict rules:
1.  No Numbers: Do not use any numbers, percentages, degrees, or other
quantitative measurements. Use descriptive words to convey scale and
magnitude (e.g., 'slightly', 'sharply', 'fully', 'gently').
2.  Be Concise: The entire instruction must be under 30 words.

The instruction should still clearly identify the component and the
action (e.g., bend, stretch, twist, inflate, rotate).
```

```
Good Example Instructions:
- "Significantly stretch both legs outwards while maintaining their
original thickness."

If there is no discernible difference between the images, respond with
the exact string: 'error'. Do not provide any other explanation.
```

## A.4 MORE QUALITATIVE RESULTS

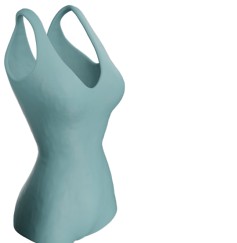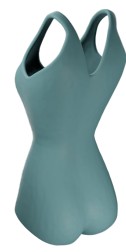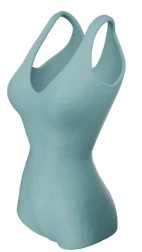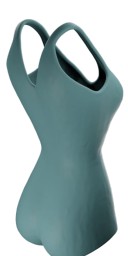

A low-top sneaker featuring a rounded toe, geometric patterns, laces, and a sole with a slightly elevated heel.

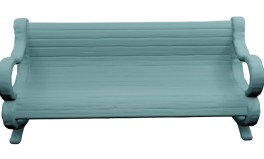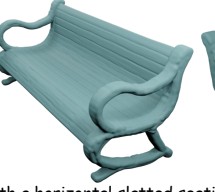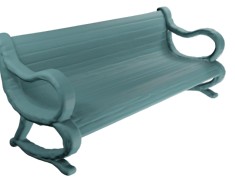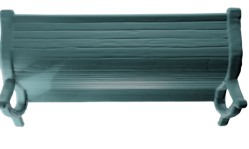

An hourglass-shaped waist trainer with vertical seams and adjustable hooks.

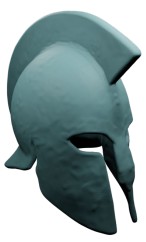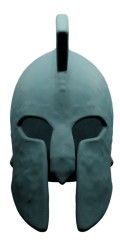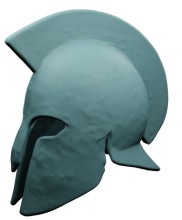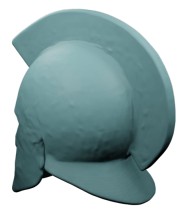

A minimalist bench with a horizontal slatted seating surface, curved legs, and scroll armrests.

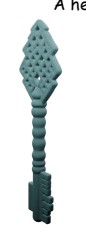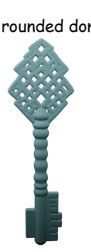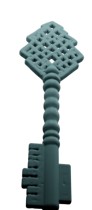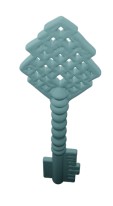

A helmet with a crest, a rounded dome shape, and ear cut-outs, inspired by Roman or Spartan styles.

A medieval key with a diamond-shaped head, an elongated cylindrical shaft, and a pointed two-pronged bit.

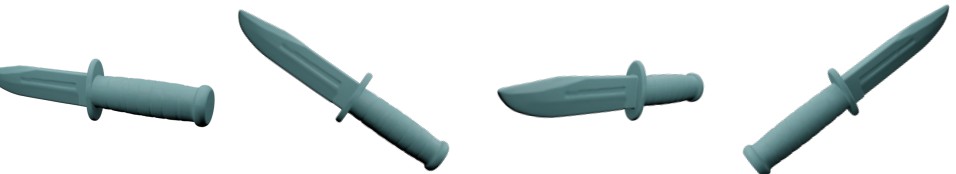

A KA-BAR knife with a sleek, long blade, a cylindrical grip featuring rings, a guard, and a pommel.

Figure 11: More text-to-3D visual results. LUCID-3D is capable of generating diverse and high-quality 3D models.

