# OpenReview forum: "LUCID-3D: A Lightweight and Compatible Framework for Unified 3D Understanding and Generation"
_ICLR.cc/2026/Conference — ICLR 2026 Conference Withdrawn Submission_

### Official Review · Reviewer_aDvx · 2025-10-25

**Soundness:** 3
**Presentation:** 3
**Contribution:** 2
**Rating:** 4
**Confidence:** 4

**Summary:**

This paper presents LUCID-3D, a unified framework that combines autoregressive and diffusion models to understand, generate, and edit 3D models. Built upon a pre-trained vision-language model (VLM), LUCID-3D incorporates an additional projector to inject 3D latent representations into the VLM. To enable generation and editing capabilities, the framework employs a diffusion model that uses latents from the VLM as conditioning input to produce 3D model representations. The authors curate two datasets using a commercial LLM: a Text-3D dataset for understanding and generation tasks, and a 3D Editing dataset for editing applications. The experiments demonstrate that LUCID-3D outperforms most existing methods in both understanding and generation tasks, while enabling precise editing of 3D shapes without requiring local region specifications.

**Strengths:**

1. It is the first unified framework to combine autoregressive model with 3D diffusion model
2. The qualitative results demonstrate outstanding performance on complex text-to-3D generation tasks.

**Weaknesses:**

1. The framework is very similar to the existing AR + diffusion frameworks in 2D generation, which somewhat limits its novelty.
2. For 3D understanding, the authors focus solely on captioning tasks. While captioning is important for aligning 3D inputs with text, it does not constitute comprehensive 3D understanding, as the system cannot answer more complex questions about 3D models.

**Questions:**

1. According to Table 1, incorporating additional 2D images significantly improves the captioning capability. This suggests that using images alone might achieve similar results. The lower performance of QwenVL-2.5 is likely because it is not fine-tuned on the image + text data from the Text-3D dataset.
2. How does the model identify which task to perform? In this paper's setting, task identification appears to rely on input types. I am curious whether any special design is made or multi-task training is included to help the model recognize the intended operation.
3. Do the 3D Generation and 3D editing use the same Diffusion Model? Dditing diffusion modes typicallyl take the latent to be edited as direct input. However, in LUCID-3D, the 3D latent is not explicitly fed to the diffusion model. It is just fused with the text and encoded into a VLM latent, which then is injected into the diffusion as condition.

---

### Official Review · Reviewer_SSTR · 2025-10-30

**Soundness:** 2
**Presentation:** 3
**Contribution:** 2
**Rating:** 4
**Confidence:** 4

**Summary:**

This paper presents LUCID-3D, a unified framework for 3D understanding and generation that combines autoregressive (AR) modeling for understanding with diffusion for generation. The approach extends a 2D VLM (QwenVL-2.5) to 3D via a lightweight MLP projector, and connects it to a 3D diffusion model (Hunyuan3D) through learnable queries and a transformer connector. The framework supports 3D captioning, text-to-3D generation, and instruction-based editing. The authors evaluate on Objaverse-based datasets and report competitive performance across tasks.

**Strengths:**

1. The framework's compatibility with different base models (demonstrated with CraftsMan3D and Hunyuan3D) is valuable for practical deployment.
2. Successfully handling understanding, generation, and editing in one framework has practical merit, even if not deeply novel.
3. The qualitative results demonstrate that the method can generate detailed, text-aligned 3D objects competitive with specialist methods.
4. Ablations on query length and latent resolution provide useful insights for practitioners.

**Weaknesses:**

1. The paper doesn't demonstrate synergy between understanding and generation. Are there emergent capabilities from joint training? Does understanding improve generation or vice versa?
2. Trellis and other baselines may be trained on different data. Fair comparison requires controlled experiments on the same training set.
3. While the paper curates new datasets, their quality relative to existing resources is not validated. The Text-3D dataset relies on GPT-4o annotations from multi-view images, which may inherit 2D biases. The editing dataset's limitation to pose changes is significant.
4. The core components (learnable queries, transformer connectors, frozen VLM approach) are directly borrowed from 2D unified models like BLIP-3 and others cited. The contribution is primarily an engineering adaptation to 3D rather than a fundamental methodological advance.
5. Current 3D model editing cannot achieve 100% success, Figure 4's qualitative comparison would benefit from showing failure cases or discussing when the method struggles. Currently only cherry-picked examples are shown.

**Questions:**

See the weaknesses. I am willing to engage in detailed discussion with the authors during the rebuttal phase.

---

### Official Review · Reviewer_6Ngp · 2025-10-31

**Soundness:** 2
**Presentation:** 3
**Contribution:** 2
**Rating:** 4
**Confidence:** 5

**Summary:**

Unlike ShapeLLM-Omni, which unifies 3D understanding and generation through a pure token-based autoregressive paradigm, LUCID-3D introduces an additional DiT module on the generation side to enhance synthesis quality, thereby supporting 3D generation, understanding, and editing in a unified framework.

**Strengths:**

1. Incorporating diffusion to improve 3D generation is a necessary step for unified 3D models, and this work represents a promising attempt in that direction.
2. Both qualitative visualizations and quantitative point-cloud results demonstrate competitive performance and potential.

**Weaknesses:**

1. The training procedure is not described in sufficient detail. I would like clarification on the specific training stages, the dataset used in each stage, the number of epochs, and which components are trainable versus frozen.
2. Compared to ShapeLLM-Omni, adding a DiT module increases training complexity. Therefore, efficiency metrics (e.g., FLOPs and other compute statistics) are needed for each stage, especially for the generation stage.
3. Evaluation on only two benchmarks is insufficient to demonstrate effectiveness and generalization. I would like to see GPT-based scoring on Objaverse captioning, results on the 3D-MMVET benchmark, and broader comparisons with more SOTA models such as ShapeLLM. For generation performance, please evaluate on the Toys4K test set using CLIP Score, KD, FD, and other standard metrics, and ensure comparisons with Trellis use the Large or larger model variants.
4.And How the version of DIT influences the generation performance, because Hunyuan3D 2.1 is very powerful base model.

**Questions:**

### **1. Training Pipeline Insufficiently Specified**

The manuscript provides only a high-level overview of the training pipeline and lacks essential reproducibility details. Please provide a more complete description including:

* **Training stages**

  * Number and purpose of each stage
  * Loss functions, optimization goals, and training objectives

* **Datasets per stage**

  * Dataset names (e.g., Objaverse, Toys4K, Hunyuan3D, proprietary data, etc.)
  * Dataset size, category coverage, and preprocessing steps

* **Training hyperparameters**

  * Epochs, batch size, learning rate schedule, weight decay, warm-up strategy
  * Gradient clipping / mixed-precision settings if applicable

* **Trainable vs. frozen components**

  * Which modules are trained or frozen at each stage (DiT, multimodal encoders, fusion modules, text decoder, etc.)
  * Rationale for freezing and unfreezing stages

* **Checkpointing and model selection**

  * Checkpoint frequency and criteria for choosing the final model

> The goal is to enable faithful reproduction of the training process and ensure scientific rigor.

---

### **2. DiT Module Introduces Additional Compute Cost — Efficiency Analysis Needed**

Compared with ShapeLLM-Omni, introducing a Diffusion Transformer (DiT) increases the computational footprint. Please include:

* **Training compute metrics**

  * FLOPs per stage, GPU memory usage, wall-clock training time
  * Throughput (samples/second) on standard GPU setups (e.g., 8×A100 / 8×4090)

* **Inference efficiency**

  * Latency and GPU memory requirements for single-sample generation
  * Any acceleration techniques used (e.g., Flash-Attention, FP8, KV cache, etc.)

* **Model size comparison**

  * Parameter counts for each module and DiT contribution

* **Compute vs. quality trade-off**

  * Comparison with ShapeLLM-Omni, Trellis, and other baselines

> This analysis is important to justify the added model complexity and verify that the performance gain is computationally reasonable.

---

### **3. Evaluation Scope Too Narrow — Broader Benchmarks Needed**

Evaluation on only two benchmarks is insufficient to demonstrate generalization and robustness. Please include:

#### **Captioning Performance**

* GPT-based scoring on **Objaverse captioning**
* Results on **3D-MMVET**, covering:

  * Textual understanding
  * 3D spatial/geometry perception
  * Cross-modal consistency

#### **Generation Performance**

* **Toys4K test set** with standard metrics:

  * CLIP Score, KD, FD, diversity, geometry quality

* Comparison against:

  * Trellis-Large (or larger variants)
  * ShapeLLM family and other strong baselines

#### **Qualitative evaluation**

* Multi-view renderings, surface topology, mesh consistency
* Failure case analysis and category-level breakdown

> A more comprehensive benchmark suite will strengthen claims of effectiveness and generalization capability.

---

### **4. Influence of DiT Variants and Hunyuan3D Base Model**

Since Hunyuan3D 2.1 provides a strong foundation, please analyze the architectural contribution of the DiT module:

* **Scaling experiments**

  * DiT-Base / DiT-Large / DiT-XL comparisons
  * Impact on CLIP Score, KD, FD, and convergence behavior

* **Scaling laws**

  * Performance vs. parameter count and compute budget

* **Ablation studies**

  * Removing or replacing the DiT module
  * Reducing depth or token count

* **Efficiency–performance trade-off**

  * Particularly for constrained compute environments

> This will clarify whether performance gains stem from DiT design, scale, or the strength of the Hunyuan3D backbone.

---

### Note · Authors · 2025-11-12

I have read and agree with the venue's withdrawal policy on behalf of myself and my co-authors.